# Nociceptor Signalling through ion Channel Regulation via GPCRs

**DOI:** 10.3390/ijms20102488

**Published:** 2019-05-20

**Authors:** Isabella Salzer, Sutirtha Ray, Klaus Schicker, Stefan Boehm

**Affiliations:** Division of Neurophysiology and Neuropharmacology, Centre for Physiology and Pharmacology, Medical University of Vienna, Waehringerstrasse 13a, A-1090 Vienna, Austria; isabella.salzer@meduniwien.ac.at (I.S.); sutirtha.ray@meduniwien.ac.at (S.R.); klaus.schicker@meduniwien.ac.at (K.S.)

**Keywords:** nociceptor, inflammatory pain, G protein-coupled receptor, voltage-gated ion channel, TRP channel, K_2*P*_ channel, Ca^2+^-activated Cl^−^ channel

## Abstract

The prime task of nociceptors is the transformation of noxious stimuli into action potentials that are propagated along the neurites of nociceptive neurons from the periphery to the spinal cord. This function of nociceptors relies on the coordinated operation of a variety of ion channels. In this review, we summarize how members of nine different families of ion channels expressed in sensory neurons contribute to nociception. Furthermore, data on 35 different types of G protein coupled receptors are presented, activation of which controls the gating of the aforementioned ion channels. These receptors are not only targeted by more than 20 separate endogenous modulators, but can also be affected by pharmacotherapeutic agents. Thereby, this review provides information on how ion channel modulation via G protein coupled receptors in nociceptors can be exploited to provide improved analgesic therapy.

## 1. Introduction

Nociception refers to *“neural processes of encoding and processing noxious stimuli”* as defined by the International Association for the Study of Pain. Noxious stimuli are *“actually or potentially tissue damaging events”* that need to act on nociceptors in order to cause pain. Accordingly, nociceptors are viewed as *“sensory receptors that are capable of transducing and encoding noxious stimuli”*. As such, nociceptors are peripheral nerve endings of first order nociceptive neurons; these are part of the peripheral nervous system with neuronal cell bodies located mostly in dorsal root ganglia and with central neurites projecting to second order nociceptive neurons located in the dorsal horn of the spinal cord [1].

Noxious stimuli that impinge on nociceptors comprise mechanical forces, temperature changes (heat and cold), and chemical agents (e.g., protons and plant-derived irritants such as capsaicin, menthol, or isothiocyanates). Apart from acting directly on nociceptors, such injurious impact may lead to inflammation, as do infections. This pathologic response is characterized by the release of a plethora of mediators from various types of cells including, amongst others, macrophages, mast cells, immune cells, platelets and the nociceptive neurons themselves [2]. Together, these mediators are called inflammatory soup and lead to an increased responsiveness of nociceptive neurons. This latter mechanism is known as sensitization and forms the pathophysiological basis of allodynia and hyperalgesia: pain in response to a non-nociceptive stimulus and increased pain sensitivity, respectively [1].

Constituents of the inflammatory soup comprise protons, nucleotides and nucleosides, enzymes (proteases), fatty acid derivatives (prostanglandins), biogenic amines (histamine, noradrenaline, and serotonin), cytokines, chemokines, neurotrophins and other peptides (bradykinin, endothelin, and tachykinins) [3]. These multifarious endogenous agents influence nociceptor signaling through a variety of different receptors:Protons act directly on ion channels that are members of either the TRP or the ASIC family [4,5].ATP as a prototypic nucleotide may activate a subset of ligand-gated ion channels known as P2X receptors [6].Cytokines such as various interleukins or tumor necrosis factors (TNFs) target different subtypes of cytokine receptors [7].Neurotrophins, in particular nerve growth factor, bind to high affinity tyrosine receptor kinases (trks) and to the low affinity receptor p75 [8].All others of the aforementioned inflammatory mediators and ATP elicit their actions on nociceptors via some type of G protein-coupled receptor (GPCR).

Hence, most of the influence of the inflammatory soup on nociceptors is mediated by GPCRs [3]. The common outcome of the separate actions of the single components contained in the inflammatory soup is sensitization of nociceptors, as mentioned above. The prime task of nociceptors is the transformation of noxious stimuli into action potentials that are propagated along the neurites of nociceptive neurons from the periphery to the spinal cord. Accordingly, sensitization means that this transformation of noxious stimuli into action potentials is facilitated, and this may occur through one of two possible mechanisms: reduction in the action potential threshold or increased responses to suprathreshold stimuli. In principle, these two pathophysiological alternatives underlie the clinical phenomena of allodynia and hyperalgesia, respectively [1]. Consequently, this review summarizes how activation of certain GPCRs can impinge on either of these two mechanisms underlying the sensitization of nociceptors.

Obviously, the transformation of noxious stimuli into action potentials relies on the coordinated operation of a variety of ion channels. Therefore, inflammatory mediators must ultimately act on the function of these ion channels to be able to sensitize nociceptors. In this regard, the present review summarizes signaling mechanisms that link an activation of GPCRs to changes in ion channel function in nociceptors.

When dealing with GPCRs expressed in peripheral nociceptive neurons, one must take into account that not all of them subserve stimulatory actions that in the end lead to sensitization. Several of these GPCRs mediate inhibitory effects which rather diminish than enhance neuronal excitability. For the sake of comprehensiveness, such inhibitory receptors are considered as well.

## 2. Ion Channels as Targets of GPCR Signaling in Peripheral Nociceptive Neurons

In this section, ion channel families are described in light of their roles in nociceptive neurons. In this respect, one can discern between ion channels that are directly involved in the sensation of noxious stimuli and those that are rather responsible for the ensuing generation and propagation of action potentials. The former group comprises TRP channels, ASICs, and mechanosensitive K${}^{+}$ and Piezo channels, whereas voltage-activated Na${}^{+}$ and Ca${}^{2+}$ channels as well as various types of K${}^{+}$ channels belong to the latter. This basic characterization of each of these ion channel families is followed by a description of the mechanisms that link activation of various GPCRs to changes in functions of these ion channels.

### 2.1. TRP Channels Involved in Pain Sensation

Transient receptor potential (TRP) channels are expressed in a variety of tissues throughout the body, such as skin, kidney, bladder, vascular smooth muscle cells and the nervous system [9]. The TRP channel family consists of six sub-families: TRPA (Ankyrin), TRPC (canonical), TRPM (melastatin), TRPML (mucolipin), TRPP (polycystin), and TRPV (vanniloid), encoded by a total of 28 genes [10]. The latter five can be further divided into subtypes: TRPC1-7, TRPM1-8, TRPML1-3, TRPP1-3, and TRPV1-6 [11]. The broad variety of TRP channels allows sensing both noxious and innocuous signals [12]. Thus far, TRPV1-4 [12] and TRPM3 [13] channels have been implicated in the sensation of noxious heat. TRPA1, TRPC5 and TRPM8 channels have been suggested to detect noxious cold temperatures [14]. In addition, both TRPA1 and TRPV4 subtypes are thought to be involved in the detection of noxious mechanical stimuli [12], while TRPA1, TRPV1, TRPV3, TRPV4, TRPM8, and TRPC3 may contribute to the sensation of itch [15,16].

The aforementioned TRP channel subtypes are expressed in different types of peripheral sensory neurons such as dorsal root ganglion (DRG) and trigeminal ganglion (TG) neurons. TRPV1 channels and TRPM8 channels are mainly expressed on separate sets of neurons. TRPV1 channels can be found in C-fibers, whereas TRPM8 channels can be found on both fiber types transferring noxious signals (Aδ- and C-fibers) [17]. Nevertheless, coexpression of TRPM8 and TRPV1 in one DRG neuron has been reported as well [18,19]. TRPA1 and TRPV1, in contrast, are mostly coexpressed in sensory neurons [17,19]. TRPV2 channels have been detected in Aδ-fibers [12]. TRPV3 channels can be found on keratinocytes rather than sensory neurons [20], and TRPV4 channels are expressed in variety of tissues including the CNS and peripheral neurons [21]. TRPM3 channels are expressed in a large subset of DRG and trigeminal ganglion neurons. TRPM3 mRNA can be detected in approximately 80% of these sensory neurons at a level that is comparable to that of TRPA1 and TRPV1. However, only small-diameter neurons produce currents mediated by TRPM3 [22].

The role of TRPV1 channels as sensors of noxious heat is well established [23]. In general, TRP channels resemble the structure of voltage-gated K${}^{+}$ (K${}_{V}$) channels. Each channel is made up of four subunits, each having six membrane spanning domains. Similar to K${}_{V}$ channels, transmembrane domains 5 and 6 comprise the channel pore, whereas transmembrane domains 1–4 resemble a voltage sensor. Both N- and C-termini are found on the intracellular side [14] and harbor a number of regulatory domains. Channel trafficking and assembly is regulated by six so-called ankyrin repeats located at the N-terminus [24,25]. TRPV1 channels are not just activated by noxious heat, but also by voltage, binding of vanilloids, such as capsaicin, or high concentrations of H${}^{+}$ ions [10]. As compared to K${}_{V}$ channels, TRPV1 channels display a rather weak voltage sensitivity [26], which can be explained by the fact that the voltage-sensing transmembrane domains 1–4 remain fairly static during activation [26,27,28]. In addition, transmembrane domain 4 of TRPV1 channels contains a lower number of positively charged amino acids as compared to K${}_{V}$ channels. Hence, an additional voltage-sensing segment might be required for TRP channels [14]. The gating in response to heat is regulated by the so-called TRP domain. However, this process remains incompletely understood [29]. The TRP domain spans 25 amino acid residues and is located immediately adjacent to transmembrane domain 6. It contains the TRP box, a stretch of conserved amino acid residues (WKFQR), which is a hallmark of TRP channels. The TRP domain is thought to be involved in a number of processes, like PIP${}_{2}$ binding or channel assembly, but the exact mechanism still needs to be fully elucidated [24]. As mentioned before, PIP${}_{2}$ is thought to regulate TRPV1 channel function, however it is still under debate if PIP${}_{2}$ is a positive or a negative regulator [30]. Cryo-EM studies in nanodiscs revealed the position of PIP${}_{2}$ in proximity to the vanilloid binding site. Binding of a vanilloid displaces a part of the PIP${}_{2}$ molecule, which reaches into the vanilloid binding pocket. The removal of the phosphoinositide is thought to lead to channel gating [26]. Such an effect would rather point towards a negative regulatory effect of PIP${}_{2}$.

The threshold value for classifying a thermal response as noxious was determined to be 43 °C [2]. TRPV1 channels activate at temperatures that exceed 43 °C, TRPV2 activate at even higher temperatures (>52 °C), whereas TRPV3 and TRPV4 channels gate in a temperature range between 26 °C and 34 °C [20]. Similar to TRPV1 channels, heterologously expressed TRPM3 channels activate at a temperature exceeding 40 °C [22]. Interestingly, mice lacking TRPV1 channels display a delayed nocifensive response only at temperatures exceeding 50 °C [31,32]. However, as compared to TRPV1 channels, the role of the other TRPV channels linked to the detection of noxious signals remains incompletely understood [12]. The role of TRPV2 to TRPV4 channels in detecting noxious signals remains debated [13], since both TRPV2 knock-out [33] and TRPV3/TRPV4 double knock-out [34] animals retain normal thermal and mechanical sensation. The nocifensive response times of TRPM3 knock-out mice is prolonged at temperatures exceeding 52 °C [22]. Mice lacking both TRPV1 and TRPM3 show a significantly increased nocifensive response time already at 45 °C. However, some sensory neurons still produce currents in response to heat. Only a triple knock-out of TRPV1, TRPM3 and, interestingly, TRPA1 leads to a complete heat-insensitivity of sensory neurons. Furthermore, these mice were completely heat insensitive in behavioral tests [35].

The detection of both noxious and innocuous cold signals is suggested to involve TRPM8 and TRPA1 channels [23]. TRPM8 channels gate at temperatures below 25 °C [20]. Knock-out animals of TRPM8 channels lose the ability to detect cool temperatures, but retain the ability to detect noxious cold signals below 15 °C [13]. Hence, the role of TRPM8 channels as cold sensors is well established, but an additional set of ion channels needs to be involved in detecting noxious cold temperatures. TRPA1 channels are thought to be involved, but their role remains controversial [20]. Rodent TRPA1 channels were found to be gated by noxious cold temperatures, however, that function is lost in primate TRPA1 channels [36]. By contrast, human TRPA1 channels, reconstituted in lipid bilayers, were found to be activated by noxious cold temperatures [37]. In addition to these conflicting results, TRPA1 channels are usually expressed on the same set of neurons as TRPV1 channels, which appears counterintuitive [17]. Furthermore, animal studies involving TRPA1 knock-out mice point towards an insignificant role in the detection of noxious cold temperatures [20]. While their role in the detection of mechanical stimuli remains controversial as well, their contribution to the detection of noxious chemical signals is well established [12]. A large number of structurally unrelated electrophilic compounds can gate TRPA1 channels [13]. These compounds covalently modify one or more of the 31 cysteine residues, which causes channel opening [38].

#### GPCR Regulation of TRP Channels

TRPV1 channels have been studied extensively for their modulation by GPCRs. Currents through TRPV1 channels are increased in response to inflammation, which mediates an enhanced depolarization and increased excitability [2]. The sensitization of TRPV1 channels can be mediated by both G$\alpha_{q}$- and G$\alpha_{s}$-coupled receptors. Stimulation of a G$\alpha_{q}$-coupled receptor leads to activation of phospholipase C (PLC), which hydrolyzes membrane bound phosphatitylinositol 1,4, bisphosphate (PIP${}_{2}$) into soluble inositol 1,4,5 trisphosphate (IP${}_{3}$) and membrane bound diacylglycerol (DAG, Figure 1). Subsequently, IP${}_{3}$ binds to IP${}_{3}$ receptors located at the membrane of the endoplasmic reticulum, which triggers the release of Ca${}^{2+}$. DAG in turn activates protein kinase C, which phosphorylates target proteins. Every step of this cascade can interfere with the function of TRPV1 channels [39]. Presence of PIP${}_{2}$ in the membrane is thought to decrease TRPV1 channel function by interfering with agonist binding [26]. If PIP${}_{2}$ is depleted from the membrane in response to PLC activation, TRPV1 activity may increase [30]. The exact role of PIP${}_{2}$ remains debated, as it was also shown to activate TRPV1 channels [30]. Activated PKC was shown to phosphorylate two serine residues at the C-terminus, which is thought to mediate sensitization [40]. A rise in cytosolic Ca${}^{2+}$ is not considered to contribute to sensitization as it usually leads to rapid channel desensitization in response to prolonged activation [39]. A large number of inflammatory modulators was shown to increase TRPV1 channels via one of these mechanisms (Table 1).

Activation of a G$\alpha_{s}$-coupled receptor stimulates the activity of adenylyl cyclase, which produces cyclic adenosine monophosphate (cAMP). This nucleotide is needed to activate protein kinase A (PKA), which then phosphorylates its target proteins. PKA-mediated phosphorylation of TRPV1 channels increases their sensitivity towards their agonists and reduces Ca${}^{2+}$-mediated desensitization [54]. Several inflammatory mediators were found to sensitize TRPV1 channels utilizing this pathway (Table 1).

By contrast, activation of a G$\alpha_{i}$-coupled receptor decreases the activity of adenylyl cyclase which reduces the abundance of cAMP and subsequent activation of PKA (Table 1). Indeed, activation of G$\alpha_{i}$-coupled cannabinoid [62,63] and μ-opioid (MOP) [60,61] receptors was shown to reduce currents of TRPV1 receptors, which is thought to contribute to the peripheral analgesic action of opioids [60,61] and cannabinoids [62,63].

In addition to the three major GPCR pathways, TRPV1 channels were shown to be sensitized by nerve growth factor (NGF), which requires the early activation of PI3 kinase and the presence of PKC and CamKII (Ca${}^{2+}$/calmodulin dependent protein kinase II) [64]. The inflammatory mediator histamine sensitizes TRPV1 channels via G$\alpha_{q}$-coupled H${}_{1}$ receptors. Instead of utilizing the signaling cascade described above, histamine-mediated sensitization requires activation of phospholipase A2 and lipoxigenases [65,66,67].

In sensory neurons, a variety of Gα${}_{i/o}$-coupled receptors were found to inhibit currents through TRPM3 channels: including GABA${}_{B}$ receptors [68,69,70], μ-opioid receptors [68,69], somatostatin receptors [68,70], CB${}_{1}$- [69], as well as, CB${}_{2}$ receptors [68], and neuropeptide Y receptors [69]. Likewise, low concentrations of noradrenaline reduce TRPM3 activity, hinting towards α${}_{2}$ as mediating receptor. However, the adrenergic receptor involved was not further characterized [68]. Whether δ-opioid receptors also mediate a TRPM3 inhibition remains controversial: while deltorphin, a δ-selective peptide was able to reduce TRPM3 function [68], the small-molecule δ-selective agonist SB205607 was not [69]. Likewise, activation of Gα${}_{i/o}$-coupled metabotropic glutamate receptors (mGluR${}_{4/6/7/8}$) did not reduce TRPM3 function [69]. In a heterologous system, Gα${}_{q/11}$-coupled M${}_{1}$ receptors were found to inhibit currents through TRPM3 channels [70]. However, in sensory neurons, activation of Gα${}_{q/11}$-coupled mGluR${}_{5}$ only weakly inhibited TRPM3 [68]. The inhibition of TRPM3 in sensory neurons involves activation pertussis toxin (PTX)-sensitive Gα${}_{i/o}$-coupled receptors [68,69,70]. The effect did not require signaling downstream of Gα${}_{i/o}$ activation, but relied on a direct interaction with the βγ dimer [68,69,70].

TRPA1 channels are sensitized by G$\alpha_{s}$- and G$\alpha_{q}$-coupled receptors in sensory neurons. A G$\alpha_{i}$-mediated interaction has not been reported for sensory neurons. Activation of G$\alpha_{q}$-coupled PAR2 receptors increased currents mediated by TRPA1 receptors in dorsal root ganglion neurons. This interaction required the activation of PLC but none of the downstream products. Consequently, depletion of PIP${}_{2}$ from the membrane was shown to be sufficient for this interaction [12]. The inflammatory mediator bradykinin was shown to increase currents through TRPA1 channels in dorsal root ganglion neurons. This effect was mediated by G$\alpha_{q}$-coupled B${}_{2}$ receptors and required the activation of PLC. Interestingly, activation of PKA was further required for the interaction of B${}_{2}$ receptors and TRPA1 channels [71]. An interaction of G$\alpha_{q}$-coupled bradykinin B${}_{1}$ receptors with TRPA1 channels was reported in behavioral experiments. This interaction relied on activation of PLC and PKC [72]. Histamine was shown to cause nocifensive behavior in a TRPA1 dependent manner. It is thought to involve G$\alpha_{q}$-coupled H${}_{1}$ receptors and activation of PLC [73]. Adenosine, another component of the inflammatory soup, was found to sensitize esophageal C-fibers and increase TRPA1 currents via G$\alpha_{s}$-coupled A${}_{2A}$ receptors. Activation of PKA is necessary for this interaction [74]. Electrophilic metabolites of prostaglandins, however, were demonstrated to activate TRPA1 channels directly [75,76].

The inflammatory mediators prostaglandin E2 (PGE${}_{2}$), bradykinin and histamine were tested for their influence on TRPM8 channel function. As opposed to the previously described members of the TRP channel family, the activity of the cool sensor TRPM8 is reduced in the presence of bradykinin and PGE${}_{2}$. However, application of histamine did not interfere with TRPM8 channel function [77]. The action of bradykinin required the mobilization of PKC [77,78], whereas PGE${}_{2}$ involved activation of PKA [77]. Bradykinin is assumed to act via G$\alpha_{q}$-coupled B${}_{2}$ receptors [79] and several modes of action have been suggested: it was found that depletion of PIP${}_{2}$ from the plasma membrane reduced heterologously expressed TRPM8 channel function [80]. However, these experiments were performed in absence of GPCRs and it remains to be established if PIP${}_{2}$ depletion is also sufficient to reduce TRPM8 channel function in a native cell system. PIP${}_{2}$ is hydrolyzed to form IP${}_{3}$ and DAG, which is required to activate PKC. PKC is thought to activate protein phosphatase I (PPI), which is suggested to dephosphorylate TRPM8 channels. The dephosphorylation of TRPM8 channels is proposed to finally inhibit TRPM8 channel function [79]. More recently, both bradykinin and histamine were found to inhibit TRPM8 channels via a direct interaction of G$\alpha_{q}$ subunits with the channels. The inhibitory effect of both mediators did not require activation of PLC or any of the subsequent steps in the signaling cascade [81]. The receptor via which PGE${}_{2}$ exerts its effect has not been determined, however it was found that activation of a G$\alpha_{s}$-coupled receptor and subsequent PKA stimulation was required. Furthermore, the exact mechanism how PKA modulates TRPM8 channel function remains unknown [79].

### 2.2. Acid-Sensing Ion Channels

Acid-sensing ion channels (ASIC) represent one of many ion channel families that detect noxious chemical stimuli. Additional chemical sensors are TRP channels, namely TRPA1 and TRPV1, as well as ATP-gated P2X receptors [82]. As the name suggests, acid sensing ion channels are activated in low pH conditions [83]. Such acidic conditions occur during an inflammatory response, ischemia, or fatiguing exercise [82]. ASICs can be divided into three subtypes ASIC1 to ASIC3. ASIC1 and ASIC2 can even be further subclassified into two splice variants each (ASIC1a, ASIC1b; ASIC2a, ASIC2b) [84]. A fourth analog, sometimes referred to as ASIC4 [85], rather affects expression levels of ASIC1a and ASIC3, instead of producing proton-gated currents [84]. The EC${}_{50}$ for proton-mediated currents via ASIC1 and ASIC3 channels ranges between a pH of 6.2 to 6.8, whereas ASIC2 channels have an EC${}_{50}$ between pH 4.1 and 5 [84]. All forms of ASICs can be detected in somata and peripheral ends of sensory neurons [85]. ASIC1a and ASIC3 channels are preferentially expressed in small diameter DRG neurons which also express TRPV1 and most likely subserve nociceptive function [86]. A functional channel is composed of three subunits and all but one subunit can participate in both homo- and heteromeric channels. Only ASIC2b does not form functional homomeric channels [87]. One subunit consists of two transmembrane domains, having both the N- and C-termini at the intracellular side [88,89]. Most of the protein is located at the extracellular side forming the large extracellular domain (termed ECD). The structure of the extracellular domain was compared to a hand holding a ball, which explains the peculiar terminology for parts of the ECD, such as palm, knuckle, thumb, finger and β-ball [90]. The ECD contains a number of regulatory domains: for example, an acidic pocket is formed by acidic amino acid residues at the subunit–subunit interface, which is involved in binding of H${}^{+}$ ions and subsequent gating [89]. ASIC channels follow a three-state kinetic model, from a closed to an activated to an inactivated state. The recovery from inactivation can only be achieved in high pH conditions [88] and this desensitized state is thought to be regulated by the thumb domain within the ECD [90].

Genetic studies have suggested that ASICs play a role in sensing mechanical signals, but the exact gating mechanism is unknown, and their role remains heavily debated [91].

#### GPCR Regulation of ASICs

A few components of the inflammatory soup have been tested for their modulatory effect on acid-sensing ion channels: histamine was shown to selectively potentiate heterologously expressed ASIC1a channels. This process involved a direct action of histamine and did not require the presence of a GPCR [92]. The nucleotides UTP and ATP were shown to increase acid-induced currents (Figure 2) in rat dorsal root ganglion neurons as well as acid-induced membrane excitability. In this respect, UTP was found to act via G$\alpha_{q}$-coupled P2Y${}_{2}$ receptors and required the activation of PLC, subsequent stimulation of PKC and the presence of the anchoring protein PICK-1 (protein interacting with C-kinase 1) [93]. A similar effect can be observed when serotonin is applied: both ASIC-mediated currents and neuronal excitability are increased [94]. Serotonin was found to act via Gα${}_{q}$-coupled 5-HT${}_{2}$ receptors [94] and required activation of PKC [94,95]. Two phosphorylation sites, one at the N-terminus and one at the C-terminus of ASIC3, need to be phosphorylated for the full effect. Again, PICK-1 is necessary for PKC-mediated phosphorylation of ASIC channels. This scaffold protein is thought to bind to ASIC2b subunits in heterotrimeric channels and to link PKC to the channel and to enable phosphorylation [95]. Another Gα${}_{q}$-coupled receptor, PAR2, was found to increase ASIC-mediated currents in rat pulmonary sensory neurons. Interestingly, neither PLC nor PKC were required for the PAR2 mediated current increase but the pathway involved was not studied further [96]. Depending on the activation mechanism of PAR2, one may observe an increase of cytosolic Ca${}^{2+}$ following the G$\alpha_{q}$-dependent activation of PLC and formation of IP${}_{3}$. On the other hand, PAR2 activation was shown to signal also via G$\alpha_{12/13}$ proteins, which activate Rho kinase and lead to ERK phosphorylation. Additionally, PAR2 activation may lead to β-arrestin recruitment. Whether PAR2 activation may also decrease or increase cAMP levels remains controversial [97].

By contrast, activation of G$\alpha_{i/o}$-coupled receptors was shown to decrease currents through ASIC channels. First, stimulation of cannabinoid CB${}_{1}$ receptors was found to reduce nocifensive behavior triggered by local acidosis which relies on an interaction between CB${}_{1}$ receptors and ASIC channels [98]. Second, activation of μ-opioid receptors was shown to decrease ASIC-mediated currents, neuronal excitability in dorsal root ganglion neurons and nocifensive behavior induced by local acidification [99]. In addition, nocifensive behavior provoked by mechanical stimuli [100] or local acidification [101] is reduced by local application of oxytocin. Oxytocin reduces ASIC-mediated currents in dorsal root ganglion neurons via activation of vasopressin V${}_{1A}$ receptors. This effect was found to be dependent on G$\alpha_{q}$ activation, but further steps of the cascade were not tested [101]. It remains to be determined if the observed differences in G$\alpha_{q}$-mediated effects on ASIC channels, for example, depend on the recruitment of PICK-1.

### 2.3. Mechanosensitive Channels in Pain Sensation

The variety of mechanical stimuli detected by so-called mechanosensors in the sensory nervous system ranges from light to noxious mechanical stimuli. These specialized neurons express mechanotransducer channels [102]. To sense noxious mechanical stimuli, high-threshold mechanosensors are required, which should express ion channels that open in response to strong mechanical stimuli and lead to a depolarization of these neurons [103]. Acid sensing ion channels (as described above) were suggested to act as mechanotransducer channels in *C. elegans*. However, heterologously expressed mammalian ASICs do not gate in response to mechanical stimuli. Hence, such depolarizing mechanotransducer channels in nociceptive neurons remain to be identified [103].

One family of ion channels that contributes to the sensation of noxious mechanical stimuli is the family of two-pore K${}^{+}$ channels (K${}_{2P}$) [103]. The family of K${}_{2P}$ channels consist of 15 members, which usually provide so-called background or leak currents, which are the major contributors to the resting membrane potential [11]. Three members of the K${}_{2P}$ family were found to be involved in sensing noxious mechanical stimuli: K${}_{2P}$2.1 (TREK1), K${}_{2P}$4.1 (TRAAK), and K${}_{2P}$10.1 (TREK2) [104]. A functional K${}_{2P}$ channel is formed by two subunits consisting of four transmembrane segments each (TMS1–4). Both the N- and C-termini are located in the intracellular space and the linker regions between TMS1 and -2, as well as TMS3 and -4 are located inside the plasma membrane to form one selectivity filter each [105]. Accordingly, a total of eight transmembrane domains and four selectivity filter regions line the ion conduction pore. This structure of the pore is highly homologous to that of voltage-gated K${}^{+}$ channels [104]. In closed conformation, the ion conduction pathway is blocked by lipid acyl side chains and membrane stretch directly gates K${}_{2P}$ channels [104]. K${}_{2P}$2.1 and K${}_{2P}$4.1 channels are strongly expressed in small-diameter DRG neurons and only weakly expressed in medium- and large-diameter DRG neurons, whereas K${}_{2P}$10.1 channels are exclusively expressed in small-diameter neurons [106]. Interestingly, knock-out of these channels leads to an increased nocifensive response to mechanical stimuli [107]. Since all these channels are selective K${}^{+}$ channels, opening of K${}_{2P}$ channels leads to an efflux of K${}^{+}$ ions and subsequent hyperpolarization. It is thought that stretch-activation of K${}_{2P}$ channels counteracts the activation of depolarizing mechanotransducer channels and thereby finetunes the mechanically induced nociceptive signal which is transferred to the brain [102]. Hence, it is clear that these three members of the K${}_{2P}$ family contribute to the perception of noxious mechanical stimuli, but they cannot represent the primary depolarizing mechanotransducer channel [103]. Such a functional entity is rather provided by Piezo channels, in particular Piezo2, which contributes to mechanically activated currents in DRG neurons [108]. However, deletion of Piezo2 impairs touch, but sensitizes mechanical pain in mice [109]. Therefore, additional sensors of mechanical pain remain to be identified.

#### 2.3.1. GPCR Regulation of Mechanosensitive Potassium Channels

The function of mechanosensitive K${}_{2P}$ channels can be adjusted by a number of modulators like arachidonic acid, polyunsaturated fatty acids, glutamate, noradrenaline, acetylcholine, TRH [105] or serotonin [107] (Figure 3). Arachidonic acid and polyunsaturated fatty acids activate these channels directly [105]. The other modulators influence K${}_{2P}$ channel function via activation of GPCR pathways. All three major GPCR pathways affect K${}_{2P}$2.1 and K${}_{2P}$10.1 channel activity: phosphorylation of two different C-terminally located serine residues either by cAMP activated PKA or PKC leads to an inhibition of both subtypes. Phosphorylation of yet another serine residue by protein kinase G (PKG) on the other hand activates K${}_{2P}$2.1 and K${}_{2P}$10.1 channels. PKG is activated by an increase of cyclic guanosine monophosphate (cGMP) which in turn is formed by soluble guanylyl cyclase. Soluble guanylyl cyclase is directly activated by nitric oxide and does not involve activation of a GPCR [105]. In dorsal root ganglion neurons, prostaglandin F${}_{2a}$ (PGF${}_{2a}$) was shown to decrease K${}_{2P}$ mediated currents [106]. The exact coupling mechanism was not elucidated, however, PGF${}_{2a}$ is the endogenous ligand for Gα${}_{q}$-coupled FP prostanoid receptors [110] and it is likely that PKC activation is involved in this process. In addition, prostaglandin E${}_{2}$ (PGE${}_{2}$)-induced nocifensive behavior was reduced in K${}_{2P}$10.1 knock-out mice, but the mechanism of action was not investigated [111].

Activation of a K${}^{+}$ permeable ion channel leads to an efflux of K${}^{+}$ ions and a subsequent hyperpolarization. K${}_{2P}$ channels are active at resting conditions and contribute to the formation of the resting membrane potential. Inhibition of these channels leads to a depolarization and a subsequent increase in excitability [107]. Other components of the inflammatory soup were examined for their effects on K${}_{2P}$ channels: for example, serotonin was also found to inhibit K${}_{2P}$2.1 and K${}_{2P}$10.1 channels via activation of 5-HT${}_{4}$ receptors in a heterologous cell system [112]. These GPCRs are Gα${}_{s}$-coupled and activate PKA, which is thought to mediate this effect [107]. Application of UTP, which may act through Gα${}_{q}$-coupled P2Y${}_{2}$, P2Y${}_{4}$ or P2Y${}_{6}$ receptors, leads to an inhibition of K${}_{2P}$ channels in mammary epithelial cells [113]. It remains to be determined if these inflammatory mediators also interact with mechanosensitive K${}_{2P}$ channels in dorsal root ganglion neurons.

On the other hand, activation μ-opioid receptors were found to increase K${}_{2P}$2.1 currents in hippocampal astrocytes [114], in a heterologous cell system [115], and in substantia gelatinosa neurons of the spinal cord. The latter mechanism is thought to be involved in the antinociceptive actions of opioids [116]. All opioid receptors are coupled to Gα${}_{i/o}$ G-proteins, which reduce the activity of adenylyl cyclase [110]. Subsequently, less cAMP is formed, which in turn leads to reduced PKA activity and less PKA-mediated phosphorylation of K${}_{2P}$ channels [105]. Since opioid receptors are also expressed in peripheral sensory neurons [117], such an interaction between μ-opioid receptors and K${}_{2P}$ channels might also exist in peripheral sensory neurons and contribute to opioid-mediated antinociception.

#### 2.3.2. GPCR Regulation of Piezo Channels

Mechanically activated and rapidly adapting currents in DRG neurons are carried by Piezo2 channels [108] and get sensitized by the activation of B${}_{2}$ bradykinin receptors, an effect that appears to involve PKA as well as PKC [118]. Akin currents are enhanced in the presence of ATP and UTP which act most likely through an activation of P2Y${}_{2}$ receptors [119]. Likewise, mechanically activated and rapidly adapting currents in DRG neurons as well as currents in cells expressing recombinant Piezo2 channels are enhanced by intracellular GTP or GTPγS which both lead to the activation of G proteins [120]. This confirms that the function of mechanosensitive Piezo2 channels can be enhanced by inflammatory mediators acting on GPCRs. Interestingly, Piezo2 channels can be inhibited by a depletion of membrane PIP${}_{2}$, and this mechanism is believed to underlie the reduction of mechanically activated rapidly adapting currents in DRG neurons in response to an activation of TRPV1 by capsaicin [121]. Why such an effect cannot be observed in DRG neurons during the activation of Gα*q*-linked GPCRs, such as B${}_{2}$ bradykinin and P2Y${}_{2}$ receptors [118,119], remains open for future investigation.

### 2.4. Calcium-Activated Chloride Channels in Pain Sensation

Calcium-activated chloride (CaCC) channels occur in a variety of tissues. They are widely expressed in the nervous system but also other tissues like vascular smooth muscle cells. In addition, CaCCs are used as a marker for stroma tumors in the gastrointestinal tract and were initially termed DOG1 [122]. Functionally described CaCCs were linked to transmembrane proteins of unknown function 16A and B (TMEM16) [123,124,125]. Thereafter, the term anoctamin was introduced to account for its predicted eight (lat. octo) membrane spanning domains and its function as an anion channel [122,126]. The family of TMEM16 proteins consists of ten members termed TMEM16A to TMEM16K (I is left out), which correspond to anoctamin 1 to anoctamin 10 (ANO1 to ANO10) [127]. Only ANO1 and ANO2 were unequivocally identified as mediating calcium-activated anion currents, the other members were either identified as calcium-activated lipid scramblases or as dual-function scramblases/ ion channels [128,129]. The subtypes ANO1, ANO2 and ANO3 were found to be expressed in sensory neurons [130,131]. ANO1 was only detected in TRPV1 positive neurons [132], whereas about 50% of ANO3 positive neurons were also TRPV1 positive [131]. Both ANO1 [133] and ANO3 [131] are involved in nociceptive behavior in mouse models of inflammatory pain. Cyro-EM- [134,135,136] and X-ray studies [137] revealed that TMEM16 analogs actually consist of ten transmembrane domains. TMEM16A forms homodimers [138,139], where transmembrane domains 3–7 of both subunits form one separate ion conduction pathway creating a two-pore anion channel [134,135,136]. Calcium, needed for gating, binds to two regions of negatively charged amino acid residues in transmembrane domains 6–8. Subsequently, transmembrane domain 6 is displaced, which ultimately opens the channel [134,135]. In the absence of Ca${}^{2+}$, CaCC function is not altered by changes in membrane voltage within physiological limits. Only voltages exceeding 100 mV may gate CaCCs directly if Ca${}^{2+}$ is absent [140], presence of cytosolic Ca${}^{2+}$ merely shifts the voltage-dependence to more physiological levels [141]. In sensory neurons, Ca${}^{2+}$ can either rise in response to activation of a Gα${}_{q}$-coupled receptor and subsequent release from intracellular stores [142], or due to an influx of extracellular Ca${}^{2+}$ via Ca${}^{2+}$-permeable ion channels like TRPV1 channels [143], or to a lesser extend voltage-gated Ca${}^{2+}$ channels [142].

Mice lacking TRPV1 receptors, the canonical heat sensor, retain some sensitivity to noxious heat and CaCCs were suggested to fulfill that task [122]. Indeed, heterologously expressed TMEM16A channels produced currents at temperatures that exceeded 44 °C [144] and tissue specific knock-out reduced mechanically and thermally induced nocifensive behavior [132,133].

#### GPCR Regulation of Calcium-Activated Chloride Channels

As described above, there are three possible sources for Ca${}^{2+}$ to activate CaCCs. One of these possibilities is to activate a Gα${}_{q}$-coupled receptor and the subsequent signaling cascade (Figure 4). In dorsal root ganglion neurons, the inflammatory mediator bradykinin was demonstrated to increase neuronal excitability via gating of CaCCs which leads to a depolarizing Cl${}^{-}$ efflux due to comparably high intracellular Cl${}^{-}$ concentrations. The induction of Cl${}^{-}$ currents through CaCCs by bradykinin required activation of PLC, formation of IP${}_{3}$ and an increase of cytosolic Ca${}^{2+}$ levels [145]. In addition, it was also shown that activation of proteinase-activated receptor 2 (PAR2) induced currents through CaCCs in dorsal root ganglion neurons. This action is dependent on increasing levels of cytosolic Ca${}^{2+}$ and a close proximity of IP${}_{3}$ receptors, which are located on the membrane of the endoplasmic reticulum, to ANO1 channels, which are located at the plasma membrane [142]. Furthermore, expression of both TMEM16A (ANO1) and PAR2 is induced in a model of neuropathic pain and both proteins are co-expressed in the same set of dorsal root ganglion neurons [146]. The inflammatory mediator serotonin was shown to induce currents mediated via CaCCs in dorsal root ganglion neurons. Even though all three types of 5-HT${}_{2}$ receptors are expressed on small-diameter dorsal root ganglion neurons, only activation of 5-HT${}_{2C}$ receptors was able to induce such currents. Furthermore, the according increase in excitability also required activation of TRPV1 channels, which provides an additional Ca${}^{2+}$ source [44]. Other possible mediators for sensitization include nucleotides: Both, UTP and ATP were found to interact with CaCCs via activation of Gα${}_{q}$-coupled P2Y${}_{1}$ and P2Y${}_{2}$ receptors. However, this interaction was only investigated in kidney [147] and pancreatic cells [148]. It remains to be established if such an interaction also contributes to nucleotide-mediated sensitization of sensory neurons.

In animal experiments, it was determined that activation of Gα${}_{i}$ coupled cannabinoid CB${}_{1}$ receptors may contribute to peripheral antinociception via an interaction with CaCCs [149]. In addition, central antinociception via activation of Gα${}_{i}$-coupled δ-opioid (DOP) receptors may involve an interaction with CaCCs [150]. However, it remains unclear how Gα${}_{i}$-coupled receptors interfere with CaCC function.

### 2.5. Voltage-Gated Na${}^{+}$ Channels

Voltage-gated Na${}^{+}$ channels (Na${}_{V}$) are crucial not just for excitable cells like central or peripheral neurons, skeletal or cardiac muscle cells, but also occur in immune cells, which are considered non-excitable [151]. In excitable cells, the principal role of Na${}_{V}$ channels is generating action potentials. Action potentials are generated, if a sufficient number of Na${}_{V}$ channels are activated in response to a local depolarization. As opposed to local depolarizations which can only spread over a few millimeters, action potentials can travel along several meters and thus transfer the information to the central nervous system. The stronger a local depolarization is, for example in response to a noxious stimulus, the more action potentials are triggered [2]. If Na${}_{V}$ channels are rendered non-functional, action potentials cannot be evoked and information transfer to the central nervous system is stopped [152,153,154,155]. This mechanism is highlighted by patients carrying loss-of-function mutations in their Na${}_{V}$1.7 or Na${}_{V}$1.9 genes, who experience insensitivity towards painful stimuli [156].

To date, nine pore-forming α-subunits of Na${}_{V}$ channels are described, designated Na${}_{V}$1.1 to Na${}_{V}$1.9 [11]. Only Na${}_{V}$1.6 to Na${}_{V}$1.9 channels can be found in nociceptive neurons. A functional voltage-gated Na${}^{+}$ channel is formed by a pore-forming α-subunit and one additional auxiliary β-subunit, of which four (β1 to β4) have been described. An α-subunit is composed of 24 transmembrane segments, which can be grouped into four domains (DI–DIV) of six transmembrane segments each (S1–S6) [151]. The transmembrane segments S1–S4 of each domain form the voltage-sensor, whereas all four S5 and S6 segments contribute to the channel pore. The S4 segments are of particular interest as they harbor a number of positively charged amino acid residues. These residues move the entire S4 segment upwards upon depolarization and lead to the gating of these ion channels. This basic principle of activation is conserved among all members of voltage-gated ion channels [157]. Voltage-gated Na${}^{+}$ channels activate within a fraction of a millisecond and subsequently enter a fast-inactivated state. This inactivation is mediated by the intracellularly located DIII-DIV linker [151]. The subunits Na${}_{V}$1.5, Na${}_{V}$1.8 and Na${}_{V}$1.9 have a low affinity for tetrodotoxin (TTX) as it ranges from 10 to 100 μM. The affinity for all other subtypes ranges between 1 to 10 nM [11]. The former subtypes are therefore described as TTX-insensitive and this represents a simple experimental tool to distinguish between Na${}_{V}$s relevant for nociception and those that are not relevant for nociception [151]. In addition to the previously described channelopathies leading to pain-insensitive patients, the importance of voltage-gated Na${}^{+}$ channels for nociception is highlighted by the fact that the most widely used local anesthetic drug, lidocaine, leads to a use-dependent block of these channels, which prevents the propagation of painful stimuli [151].

### 2.6. GPCR Regulation of Voltage-Gated Na${}^{+}$ Channels

The activity of TTX-resistant voltage-gated Na${}^{+}$ channels is affected by activation of Gα${}_{s}$ coupled receptors (Figure 5). Direct activation of adenylyl cyclase by forskolin increases TTX-resistant Na${}^{+}$ currents in dorsal root ganglion neurons. Activation of adenylyl cyclase increases the formation of cAMP, which wash shown to be involved in this process [158]. In sensory neurons, the inflammatory mediators serotonin [158], PGE${}_{2}$ [159,160] and CGRP [161] were found increase TTX-resistant Na${}^{+}$ currents via a mechanism involving Gα${}_{s}$-coupled 5-HT${}_{4}$ [162,163], EP${}_{4}$ [164], and CGRP [161] receptors, respectively. Application of substance P leads to an increase in neuronal excitability in small diameter dorsal root ganglion neurons [165]. Substance P activates neurokinin 1 (NK${}_{1}$) receptors, amongst others [110]. These Gα${}_{s}$- and Gα${}_{q}$-coupled receptors were found to increase TTX-resistant Na${}^{+}$ currents in dorsal root ganglion neurons in a PKCϵ dependent manner [166]. In addition, ATP, another component of the inflammatory soup, was also found to increase TTX-resistant Na${}^{+}$ currents in sensory neurons. By contrast, TTX-sensitive currents were not affected by application of ATP [167]. The underlying signaling cascade was not elucidated and hence it remains to be determined if, for example, Gα${}_{s}$-coupled P2Y${}_{11}$ receptors could be involved.

On the other hand, activation of protease activated receptor 2 (PAR2) does not affect TTX-resistant Na${}^{+}$ currents in nociceptive neurons [168]. These GPCRs are coupled to heterotrimeric Gα${}_{q}$-proteins and lead to release of Ca${}^{2+}$ from intracellular stores as well as activation of PKC [142].

With respect to Gα${}_{i/o}$-coupled receptors, only activation of μ-opioid receptors by the agonist DAMGO was investigated. In sensory neurons, application of DAMGO was able to prevent the PGE${}_{2}$-mediated increase of TTX-resistant Na${}^{+}$ channels [169]. The involved pathway was not investigated, but is seems reasonable to assume an interference of the Gα${}_{i/o}$-pathway with the Gα${}_{s}$-mediated potentiation of these NaV channels.

### 2.7. Voltage-Gated Ca${}^{2+}$ Channels

Neuronal calcium channels are protein complexes formed by a pore-forming α1 subunit, one β and one α2δ subunit, the latter regulating membrane trafficking and voltage dependence [170]. There are ten different α1 subunits that can be divided into three families (Ca${}_{V}$1.x–3.x). Each of these families consists of several subtypes that differ in biophysical parameters, expression pattern and physiological function. In addition, there is a clear distinction in the voltage dependence of Ca${}_{V}$1.x and Ca${}_{V}$2.x when compared to Ca${}_{V}$3.x. While the latter activates at quite hyperpolarized potentials (<−50 mV), thus being termed low voltage activated channel (LVA), the former need much higher depolarizations to be opened and are called high voltage activated channels (HVA). In DRG neurons, all three families of Ca${}_{V}$s are found [171] and all can be modulated by GPCRs to different extents. Depending on the type of GPCR, the channels are modulated via different pathways.

#### GPCR Regulation of Voltage-Gated Ca${}^{2+}$ Channels

The most prominent and by far best studied mechanism is the so-called voltage-dependent inhibition (Figure 6). This is found in Ca${}_{V}$2.x channels, where the G-protein Gβγ subunits can directly bind to the calcium channel α1 subunit. This binding event leads to a shift in the gating mode from *“wiilling”* to a *“reluctant”* one that manifests itself mainly in a marked slowing of activation [172,173,174]. The term “voltage dependent” refers to the fact that depolarization can relief the channels from inhibition and restore normal gating. As this type of inhibition is mostly exerted by Gα${}_{i/o}$-coupled receptors it can be abolished by treating the cells with pertussis toxin (PTX) that ribosilates Gα${}_{i/o}$-proteins and renders them inactive.

One of the best studied examples for voltage dependent calcium channel inhibition is the action of opioid receptors. All three types of opioid receptors, μ (MOP), κ (KOP) and δ (DOP), are found in DRG neurons, with the exact expression pattern depending on the cell type [175,176]. Initially, it was found that exposure of nociceptive neurons to opioids leads to a shortening of action potential durations [177,178,179,180,181]. It was demonstrated that application of [D-Ala2]-enkephalin (DADLE), an unspecific opioid receptor agonist, not only reduced action potential duration, but also diminished substance P release in these neurons [177]. As became clear later on, both effects were caused mainly by the inhibition of Ca${}^{2+}$ channels [172,182]. The major target for opioid modulation are Ca${}_{V}$2.x channels [183,184,185,186,187,188]. These channels are found at the presynapse and govern neurotransmitter release [170], thus inhibition of Ca${}_{V}$2.x channels leads to reduced Ca${}^{2+}$ influx and concomitantly reduced transmitter release from peripheral nociceptive neurons onto second-order neurons of the pain pathway located in the spinal dorsal horn. In line with the fact that opioid receptors couple predominantly to Gα${}_{i/o}$ G proteins [110], opioid induced calcium channel inhibition was found to be PTX sensitive [188], and voltage dependent [172]. These findings were corroborated by intracellular administration of Gα${}_{o}$ antiserum, that strongly reduced opioid receptor mediated I${}_{Ca}$ inhibition [189], unequivocally demonstrating the mechanism of action.

Besides opioid receptors, many other GPCRs expressed in DRG neurons were found to lead to a similar kind of calcium channel modulation. For example GABA${}_{B}$ [190], adenosine A${}_{1}$ [191], 5-HT [163,192,193,194,195], P2Y [196], cannabinoid [197], neuropeptide Y Y${}_{2}$ receptors [198,199,200], somatostatin SST${}_{4}$ [201] or α${}_{2}$ adrenoceptors [202].

Besides the well-studied Gβγ mediated voltage dependent inhibition, other mechanisms have also been described [174]. Most prominently, phospholipase C (PLC) mediated PIP${}_{2}$ depletion can lead to inhibition of calcium channels [203,204,205]. Similar to K${}_{V}$7 and K${}_{ir}$3 channels (see below), PIP${}_{2}$ stabilizes the open state of calcium channels. Thus, a reduction in membrane PIP${}_{2}$ leads to a voltage independent decrease in channel open probability.

While several reports about this kind of inhibition exist from sympathetic neurons [174], there are few data from nociceptive neurons. However, given the similarity of receptor and channel expression between sympathetic and sensory neurons, it is reasonable to assume that these findings will also hold true in DRG neurons. Only recently, the Mas-related G protein coupled receptor type C (MrgC) was found to inhibit high voltage-gated calcium channels in a PLC dependent manner [206].

Recently, a potentially novel form of calcium channel inhibition has been described. Huang et al. [207] found that GABA${}_{B}$ receptors not only inhibit HVA channels but also LVA channels. Inhibition of both channel types was PTX sensitive, however the LVA inhibition was strongly reduced by application of DTT, pointing to a novel mechanism that involves redox processes.

GPCRs cannot only inhibit Ca${}^{2+}$ channels but they are also known to be able to facilitate their function. A classic example would be PKA which phosphorylates Ca${}_{V}$1.x channels and increases their currents [208]. In line with this, a broadening of the action potential upon application of noradrenaline to DRG neurons was reported. This increase in action potential duration could finally be attributed to an increase in Ca${}_{V}$1.x currents [202].

### 2.8. Voltage-Gated K${}^{+}$ Channels

#### 2.8.1. K${}_{V}$7 Channels

Amongst potassium channels, K${}_{V}$ channels constitute the most diverse group with 12 known families [209]. Literature abounds on how various K${}_{V}$ channels modulate nociception at different levels of the pain pathway [210]. The prototypical structural assembly of K${}_{V}$ channels involves six transmembrane segments of which the first four (S1-S4) constitute the voltage-sensing domain (VSD) while the S5 and S6 segments constitute the pore through which K${}^{+}$ ions pass [211,212]. Amongst others, the K${}_{V}$7 channel family has received immense attention in lieu of its amenability by GPCR modulation [213,214], with five known members (K${}_{V}$7.1– K${}_{V}$7.5) encoded by KCNQ1-5 genes [209]. Four monomers come together in a homo- or heterotetrameric configuration in a subunit-specific way to yield a functional K${}_{V}$7 channel [215]. The electrophysiological correlate of K${}_{V}$7 channel activity is a slowly deactivating, non-inactivating current that has an activation threshold below −60 mV. This conductance is also known as M current as it was originally described as a current that is suppressed by an activation of muscarinic acetylcholine receptors [216]. In nociceptors, these channels contribute majorly to the resting membrane potential [217]. K${}_{V}$7 channels are expressed in all functional parts of a first-order neuron which include free nerve endings, nodes of Ranvier, and the somata of dorsal root ganglion (DRG) neurons [218]. They regulate action potential (AP) firing, which is the basis for encoding of noxious stimuli in the pain pathway [219]. Enhancing K${}_{V}$7 currents exerts analgesic effects since hyperpolarizing the resting membrane potential of nociceptors decreases neuronal excitability [220,221]. Similarly, inhibiting K${}_{V}$7 currents is proalgesic since concomitant depolarization of the resting membrane potential enhances neuronal firing [222,223].

#### 2.8.2. GPCR Regulation of K${}_{V}$7 Channels

A plethora of neurotransmitters and neuropeptides modulate K${}_{V}$7 channel function via GPCR signaling (Figure 7), specifically of the Gα${}_{q/11}$ class [224]. One of the early reports was of the nociception-relevant neuropeptide Substance P (SP) that inhibited K${}_{V}$7 currents in bullfrog DRG neurons [225]. Subsequently it was revealed that neurokinin A (NKA inhibited currents through K${}_{V}$7 channels in bullfrog DRG neuronsvia NK${}_{1}$ receptors which were coupled to PTX-insensitive G proteins [226], even though this receptor may impinge on the functions of K${}_{V}$7 channels through G protein-independent mechanisms as well [227]. The activation of Gα${}_{q/11}$-coupled receptors leads phospholipase Cβ (PLCβ) to hydrolyze the membrane phospholipid phosphatidylinositol 4,5-bisphosphate (PIP${}_{2}$) into inositol-1,4,5-trisphosphate (IP${}_{3}$) and diacylglycerol (DAG) [228]. The function of K${}_{V}$7 channels is governed by the presence of sufficient membrane PIP${}_{2}$ pools and depletion of membrane PIP${}_{2}$ levels leads K${}_{V}$7 channels to close [229]. The proalgesic mediator bradykinin mediates inhibition of K${}_{V}$7 currents via its actions on the B${}_{2}$ receptor, a Gα${}_{q/11}$-coupled receptor [230]. The active nociception mediated by bradykinin, consequent to enhanced neuronal excitability can be attenuated by prior application of the K${}_{V}$7 channel opener retigabine [145]. In addition to membrane PIP${}_{2}$ depletion, the inhibition of K${}_{V}$7 channels via IP${}_{3}$-mediated increase in intracellular Ca${}^{2+}$ levels and subsequent binding to calmodulin is well known [231,232]. In sympathetic neurons, the governing factor for substrate (PIP${}_{2}$)- versus product (Ca${}^{2+}$)-mediated inhibition consequent to application of bradykinin is contingent upon Ca${}^{2+}$ availability and rate of PIP${}_{2}$ synthesis [233]. B${}_{2}$ receptors are closely opposed to the endoplasmic reticulum (ER) where IP${}_{3}$ can diffuse and consequently mobilize Ca${}^{2+}$ reserves [216]. In DRG neurons, bradykinin primarily employs the Ca${}^{2+}$ axis to inhibit K${}_{V}$7 currents as evidenced by the fact that inhibition of Ca${}^{2+}$ release from IP${}_{3}$-sensitive stores with pharmacological tools as well as chelation of intracellular Ca${}^{2+}$ prevents bradykinin-mediated inhibition of K${}_{V}$7 channels, akin to direct activation of PLC [145]. One of the targets of the inflammatory soup is the protease-activated receptor 2 (PAR2), a Gα${}_{q/11}$-coupled receptor expressed in nociceptors [49,234]. Activation of these receptors has an inhibitory impact on K${}_{V}$7 currents leading to nociception which requires concurrent increase in cytosolic Ca${}^{2+}$ levels in addition to depletion of PIP${}_{2}$ levels [235]. Another example is the modulation of excitability in nociceptors by nucleotides. The P2Y${}_{1}$ and P2Y${}_{2}$ receptors are Gα${}_{q/11}$-coupled receptors [236]. Activation of these receptors by the nucleotides adenosine diphosphate (ADP), and 2-thio-uridine triphosphate (2-thio-UTP), respectively, leads to the inhibition of currents through K${}_{V}$7 channels [47]. Moreover, the observed effects were prevented by the application of U73122, a PLC inhibitor, inhibition of Ca${}^{2+}$ATPases by thapsigargin, and chelation of intracellular Ca${}^{2+}$ levels by BAPTA-AM [47].

The GPCRs encoded by Mas-related genes (Mrgs) are a more recently identified subset of GPCRs that are widely expressed in sensory neurons and implicated in the modulation of nociceptive information [237,238]. Specifically, the MrgD isoform is expressed in DRG neurons especially in non-peptidergic, small-diameter IB4-postive C-fiber somata [239,240]. Activation of endogenous MrgD with the agonist alanine results in the inhibition of K${}_{V}$7 currents in DRG neurons, mainly employing a pertussis toxin-sensitive pathway implicating the involvement of Gα${}_{i/o}$. Such an inhibition translates into enhanced neuronal firing in phasic DRG neurons, which classically shoot single single APs [241]. Recombinant cell-lines coexpressing MrgD receptors and K${}_{V}$7.2/7.3 heteromers exhibit an inhibition of K${}_{V}$7 currents upon stimulation with alanine, an effect that could be reversed partially by pharmacologically blocking Gα${}_{i/o}$ and reversed completely by PLC inhibition [241].

#### 2.8.3. K${}_{V}$1.4, K${}_{V}$3.4 and K${}_{V}$4 Channels

The K${}_{V}$ channels K${}_{V}$1.4, K${}_{V}$3.4 and K${}_{V}$4 members contribute to the so-called transient A-current (I${}_{A}$) [242,243], which plays a key role in regulating AP firing in DRG neurons [210,244]. The somata, axons and central terminals of DRG neurons that abut the spinal dorsal horn are enriched with K${}_{V}$3.4 channels. The expression of K${}_{V}$4.3 channels, on the other hand is restricted to the somata of non-peptidergic DRG neurons [245]. The rapidly inactivating K${}_{V}$3.4 channel is a key player in AP repolarization in DRG neurons [246,247]. Activating PKC through physiological or pharmacological means leads to a decline in fast N-type inactivation in endogenously expressed K${}_{V}$3.4 channels. This directly impacts the biophysical properties of the nociceptor: the AP gets narrowed while the AP repolarization is accelerated [248]. Specific siRNAs that selectively target K${}_{V}$3.4 channel expression abolish the changes in AP waveform mediated by PKC activation [248]. In a rat model of cervical spinal cord injury (SCI), the surface expression of K${}_{V}$3.4 channels was shown to be impaired; such dysregulation was associated with the failure of PKC to shorten the AP duration in DRG neurons [249]. Similarly, the phosphatase calcineurin (CaN) antagonizes PKC activity as revealed by a reduction in the inactivation of K${}_{V}$3.4 channels upon pharmacological inhibition of the former [250].

#### 2.8.4. GPCR Regulation of K${}_{V}$1.4, K${}_{V}$3.4 and K${}_{V}$4 Channels

Neuromedin U (NMU) is a neuropeptide that decreases neuronal excitability in DRG neurons via the enhancement of I${}_{A}$ currents through its actions on the NMU type 1 receptor (NMUR1, Figure 8) [251]. NMUR1 couples to Gα${}_{o}$ proteins, PKA and the ERK pathway in a sequential manner [251]. On the other hand, the cyclic undecapeptide urotensin-II activates the urotensin-II receptor (UTR), which couples to Gα${}_{q/11}$ [252,253]. This leads to a reduction in I${}_{A}$ in a dose-dependent fashion in trigeminal ganglion neurons, mediated via the activation of PKC [254]. The concomitant recruitment of ERK signaling cascade culminates in an enhanced excitability of TG neurons [254]. In rat TG neurons, the K${}_{V}$1.4, K${}_{V}$3.4, K${}_{V}$4.2 and K${}_{V}$4.3 channels are co-expressed with P2Y${}_{2}$ receptors [255], which in turn couple to different G proteins [256]. The application of UTP, an agonist at the P2Y${}_{2}$ receptor inhibits I${}_{A}$ currents via the ERK pathway and enhances excitability in these neurons, an effect that can be reversed by the P2Y${}_{2}$ receptor antagonist suramin [255].

### 2.9. G-Protein Activated, Inwardly Rectifying Potassium Channels

G protein activated, inwardly rectifying potassium channels (GIRK) are homo- or heterotetrameric channels formed from four different subunits (K${}_{ir}$3.1–3.4) encoded by the genes KCNJ3, KCNJ6, KCNJ9 and KCNJ5, respectively [257,258]. They are activated by pertussis toxin sensitive G proteins via binding of βγ dimers to the channel [257]. In addition, it has also been demonstrated that the Gα subunit can directly bind to the channels modulating their basal activity in the absence of GPCR activation [259]. Besides their G protein-mediated modulation, it has been demonstrated several times that K${}_{ir}$3 channels bind PIP${}_{2}$ and that this is necessary for their function [260,261].

#### GPCR Modulation of Girk Channels

Compared to their role in the CNS, data on their physiological role in DRG neurons is scarce [258]. In rat DRG neurons, co-localization of GIRK channels and μ-opioid receptors has been found [262]. The picture is complicated by the fact that, while all four subunits are expressed in rat and human DRG neurons [117,263], only low mRNA levels and a lack of immunostaining have been reported in mice [117]. This is corroborated by the fact that local application of DAMGO (Figure 9), a μ-opioid receptor agonist, is ineffective in inflammatory pain mouse models [117]. This lack of effect, however, could be overcome by expressing K${}_{ir}$3.2 in Na${}_{V}$1.8 expressing nociceptive neurons [117], demonstrating the importance of K${}_{ir}$3 channels for peripheral analgesia. These findings are contrasted by the fact that GIRK currents could be induced in a small number of about 15–20% of mouse DRG neurons by application of DAMGO [264], rendering the interpretation of mouse data difficult.

## 3. Conclusions

In this review, members of ten different ion channel families that are expressed in sensory neurons are described with respect to their contribution to nociception. Appropriate gating of these channels is subject to modulation by at least 35 different types of GPCRs, which are targeted by more than 20 separate endogenous modulators (Table 2). Thereby, GPCRs provide the largest superfamily of receptors, activation of which can mediate pro- as well as antinociceptive effects. Accordingly, these GPCRs are relevant as potential targets for analgesic drugs. However, only a few of them are currently exploited in analgesic therapy, such as opioid, cannabinoid, and CGRP receptors or prostanoid receptors as indirect targets of cyclooxygenase inhibitors. Therefore, this review should be viewed as incitement to further investigate how the modulation of ion channels via GPCRs might be tackled to provide novel pharmacotherapeutic agents for improved analgesic therapy.

## Figures and Tables

**Figure 1 ijms-20-02488-f001:**
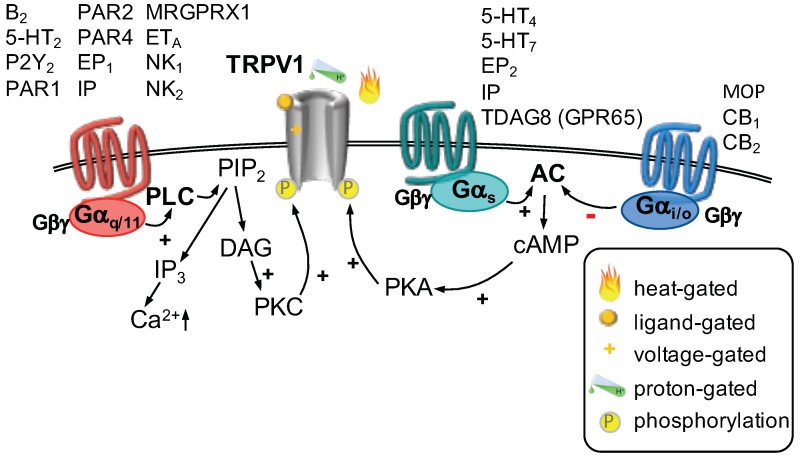
TRPV1 channels can be gated by different mechanisms (as indicated). Three major G-protein-dependent pathways modulate the function of TRPV1 channels. Activation of Gα${}_{q/11}$-coupled receptors (**left**) leads to activation of phospholipase C (PLC), which hydrolyzes phosphatitylinositol 1,4, bisphosphate (PIP${}_{2}$) into inositiol 1,4,5 trisphosphate (IP${}_{3}$) and diacylglycerol (DAG). DAG activates protein kinase C (PKC), which phosphorylates TRPV1 channels, thereby increasing their function. Activation of a Gα${}_{s}$-coupled receptor (**center**) stimulates adenylyl cyclase (AC), which produces cyclic adenosine monophosphate (cAMP). Subsequent activation of protein kinase A (PKA) leads to phosphorylation of TRPV1 channels and an increase in current. Stimulation of a Gα${}_{i/o}$-coupled receptor (**right**) decreases AC activity. Therefore, less cAMP is formed, PKA is less active and hence TRPV1 channels are not phosphorylated which decreases their activity.

**Figure 2 ijms-20-02488-f002:**
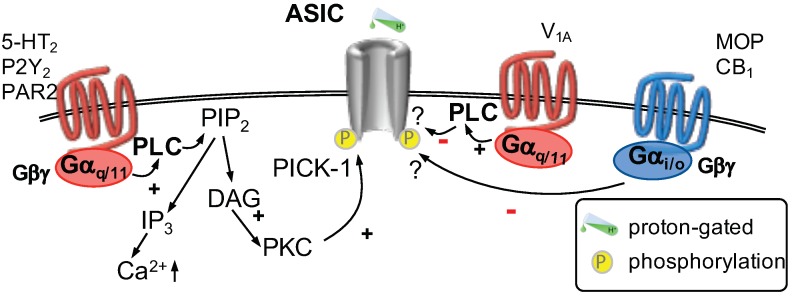
Acid sensing ion channels (ASIC) are gated by increasing concentrations of H${}^{+}$. Two G-protein pathways modulate the function of ASICs. Activation of Gα${}_{q/11}$-coupled receptors (**left**) leads to activation of phospholipase C (PLC) which hydrolyzes phosphatitylinositol 1,4, bisphosphate (PIP${}_{2}$) into inositiol 1,4,5 trisphosphate (IP${}_{3}$) and diacylglycerol (DAG). DAG activates protein kinase C (PKC) which phosphorylates ASICs increasing their function. The interaction of PKC with ASICs requires the presence of PICK-1 (protein interacting with C-kinase 1). By contrast, activation of Gα${}_{q/11}$-coupled V${}_{1A}$ receptors (**center**) were shown to decrease ASIC-mediated currents via an unknown mechanism. Stimulation of Gα${}_{i/o}$-coupled receptors (**right**) was shown to decrease currents through ASICs involving an undetermined mechanism.

**Figure 3 ijms-20-02488-f003:**
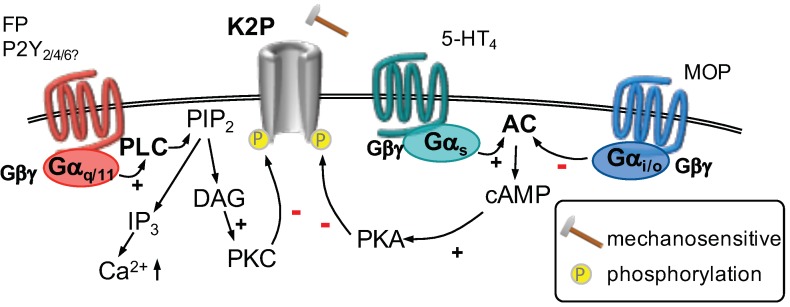
K${}_{2P}$ channels are opened in response to mechanical stimuli. Stimulation of a Gα${}_{q/11}$-coupled receptor (**left**) activates phospholipase C (PLC). Hydrolyzation of phosphatitylinositol 1,4, bisphosphate (PIP${}_{2}$) forms inositiol 1,4,5 trisphosphate (IP${}_{3}$) and diacylglycerol (DAG). DAG activates protein kinase C (PKC) which phosphorylates K${}_{2P}$ channels decreasing their function. Activation of Gα${}_{s}$-coupled receptors (**center**) leads to activation of adenylyl cyclase (AC) which produces cyclic adenosine monophosphate (cAMP). Thereafter, protein kinase A (PKA) is activated which phosphorylates K${}_{2P}$ channels and decreases their currents. Stimulation of Gα${}_{i/o}$-coupled receptors (**right**) decreases AC activity. Therefore, less cAMP is formed, PKA is less active and subsequently K${}_{2P}$ channels are not phosphorylated, which increases their activity.

**Figure 4 ijms-20-02488-f004:**
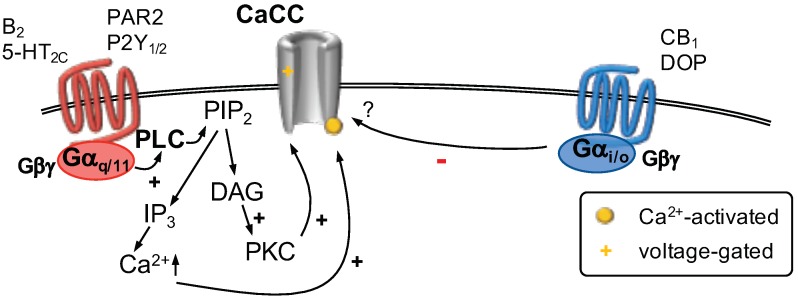
Calcium-activated Cl${}^{-}$ channels (CaCC) are gated by increasing concentrations of cytosolic Ca${}^{2+}$ and influenced by membrane voltage. The source of Ca${}^{2+}$ may be Ca${}^{2+}$-permeable ion channels located in proximity to CaCCs (not shown) or Ca${}^{2+}$ released from intracellular stores. Stimulation of Gα${}_{q/11}$-coupled receptors (**left**) activates phospholipase C (PLC). The subsequent hydrolysis of phosphatitylinositol 1,4, bisphosphate (PIP${}_{2}$) forms inositiol 1,4,5 trisphosphate (IP${}_{3}$) and diacylglycerol (DAG). DAG activates protein kinase C (PKC) which influences CaCC function. IP${}_{3}$ binds to IP${}_{3}$ receptors located in the membrane of the endoplasmic reticulum (ER) which causes Ca${}^{2+}$ release from the ER. Activation of Gα${}_{i/o}$-coupled receptors (**right**) was shown to decrease CaCC currents via an unknown mechanism.

**Figure 5 ijms-20-02488-f005:**
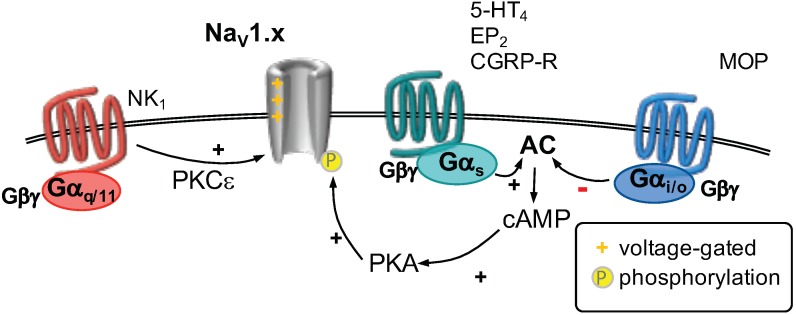
Na${}_{V}$1.x channels are activated by depolarizing voltages (as indicated). Activation of Gα${}_{s}$-coupled receptors stimulates adenylyl cyclase (AC) activity (**center**). Subsequently, cyclic adenosine monophosphate (cAMP) is formed, which activates protein kinase A (PKA). PKA-mediated phosphorylation of voltage-gated Na${}^{+}$ channels increases their currents. By contrast, activation of Gα${}_{i/o}$-coupled receptors (**right**) decreases AC activity and counteracts Gα${}_{s}$-mediated current increases. Activation of a Gα${}_{q/11}$-receptor (**left**)was shown to increase Na${}_{V}$-mediated currents involving protein kinase C ϵ (PKCϵ).

**Figure 6 ijms-20-02488-f006:**
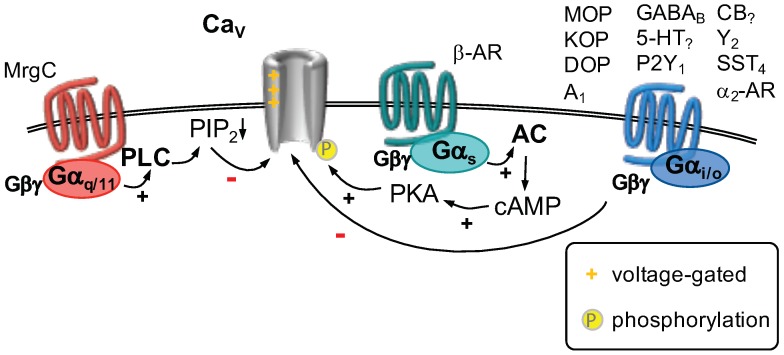
Ca${}_{V}$ channels are activated by depolarizing voltages (as indicated). Activation of a Gα${}_{i/o}$-coupled receptor (**right**) leads to a phenomenon called “voltage-dependent inhibition”. This involves direct binding of the Gβγ dimer to Ca${}_{V}$2.x channels. Activation of Gα${}_{s}$-coupled receptors (**center**) leads to activation of adenylyl cyclase (AC), which forms cAMP (cyclic adenosine monophosphate) to activate protein kinase A (PKA). PKA-mediated phosphorylation of Ca${}_{V}$1.x channels was shown to increase their currents. Activation of a Gα${}_{q/11}$-coupled receptor (**left**) stimulates phospholipase C (PLC), which hydrolyzes phosphatitylinositol 4,5 bisphosphate (PIP${}_{2}$). Depletion of PIP${}_{2}$ is sufficient to decrease currents through Ca${}_{V}$ channels.

**Figure 7 ijms-20-02488-f007:**
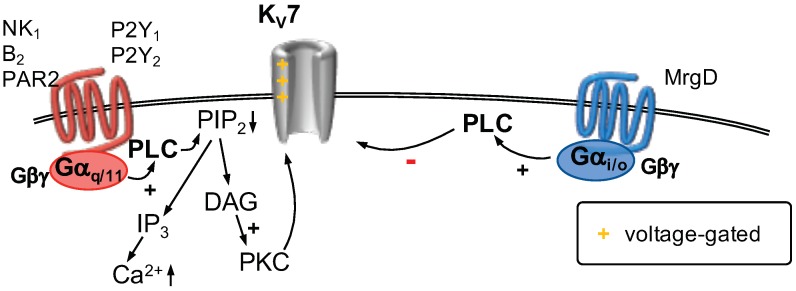
K${}_{V}$7 channels are activated by depolarizing voltages (as indicated). Gα${}_{q/11}$-coupled receptors (**left**) activate phospholipase C (PLC), which cleaves phosphatitylinositol 4,5, bisphosphate (PIP${}_{2}$) into inositol 1,4,5 trisphosphate (IP${}_{3}$) and diacylglycerol (DAG). Depletion of PIP${}_{2}$ from the plasma membrane is sufficient to decrease currents through K${}_{V}$7 channels. In addition, Ca${}^{2+}$ is released from the endoplasmic reticulum subsequent to formation of IP${}_{3}$. Ca${}^{2+}$ decreases K${}_{V}$7 currents via an interaction with calmodulin. Activation of a Gα${}_{i/o}$-coupled receptor (**right**) was shown to decrease K${}_{V}$7 currents in a PLC-dependent manner.

**Figure 8 ijms-20-02488-f008:**
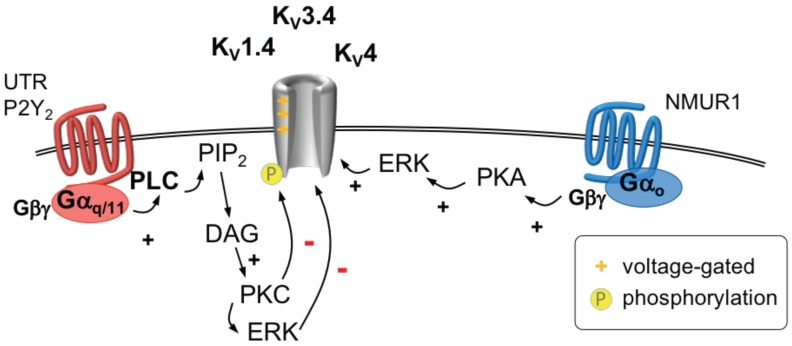
The so-called A-current is mediated by voltage-activated K${}_{V}$1.4, K${}_{V}$3.4, and K${}_{V}$4 channels. Activation of a Gα${}_{o}$-coupled receptor (**right**) increases A-type currents via a mechanism involving the Gβγ dimer, protein kinase A (PKA) and extracellular signal-regulated kinase (ERK). Stimulation of Gα${}_{q/11}$-coupled receptors (**left**) activates phospholipase C (PLC) leading to hydrolysis of phosphatityl 4,5 bisphosphate (PIP${}_{2}$) to diacylglycerol (DAG) and IP${}_{3}$. DAG activates protein kinase C (PKC), which phosphorylates A-type channels and thus inhibits these channels. ERK is activated in parallel which also phosphorylates A-type channels and thereby decreases their function.

**Figure 9 ijms-20-02488-f009:**
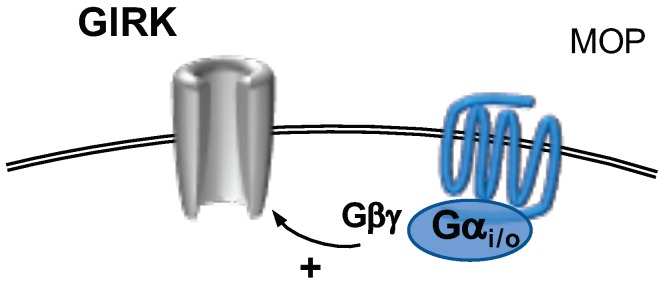
G-protein activated, inwardly rectifying K${}^{+}$ channels (GIRK) are activated subsequent to stimulation of Gα${}_{i/o}$-coupled receptors. The dissociated Gβγ dimer binds directly to GIRK channels.

**Table 1 ijms-20-02488-t001:** GPCRs modulating TRPV1 function.

GPCR Ligand	Involved GPCR	Pathway	Effect on TRPV1	Reference
Bradykinin	B${}_{2}$	G$\alpha_{q}$-DAG-PKC	increased current	[2,41]
Serotonin	5-HT${}_{2}$	G$\alpha_{q}$-DAG-PKC	increased current	[42,43,44]
	5-HT${}_{4}$	G$\alpha_{s}$-AC-PKA	increased current	[43]
	5-HT${}_{7}$	G$\alpha_{s}$-AC-PKA	increased current	[42]
UTP	P2Y${}_{2}$	G$\alpha_{q}$-DAG-PKC	increased current	[45,46,47]
BAM 8-22	MRGPRX1	G$\alpha_{q}$-DAG-PKC-PIP${}_{2}$	increased current	[48]
Proteases	PAR2	G$\alpha_{q}$-PKC	increased current	[49,50]
	PAR1	G$\alpha_{q}$-PKC	increased current	[51]
	PAR4	G$\alpha_{q}$-PKC	increased current	[51]
PGE${}_{2}$	EP${}_{1}$	G$\alpha_{q}$-PKC	increased current	[52]
	EP${}_{2}$	G$\alpha_{s}$-PKA	increased current	[52,53,54]
PGI${}_{2}$	IP	G$\alpha_{q}$-PKC	increased current	[52]
	IP	G$\alpha_{s}$-PKA	increased current	[52]
Endothelin-1	ET${}_{A}$	G$\alpha_{q}$ -PKC	increased current	[55,56]
Substance P	NK${}_{1}$	G$\alpha_{q}$ -PKCϵ	increased current	[57]
	NK${}_{2}$	G$\alpha_{q}$-PKC	increased current	[58]
H${}^{+}$	TDAG8 (GPR65)	G$\alpha_{s}$-PKA	increased current	[59]
Morphine	MOP	G$\alpha_{i}$-reduced AC	decreased current	[60,61]
Endocannabinoids	CB${}_{1}$	G$\alpha_{i}$-reduced AC	decreased current	[62]
	CB${}_{2}$	G$\alpha_{i}$-reduced AC	decreased current	[63]

AC, adenylyl cylcase; DAG, diacylglycerol; PKA, proteinkinase A; PKC, protein kinase C; BAM 8–22, bovine adrenal medulla peptide 8–22.

**Table 2 ijms-20-02488-t002:** Endogenous ligands for GPCRs modulating ion channel function.

Endogenous Ligand	GPCR	ASIC	CaCC	Ca${}_{V}$	GIRK	K${}_{2}$P	K${}_{V}$1.4 K${}_{V}$3.4 K${}_{V}$4	K${}_{V}$7	Na${}_{V}$	TRPA1 TRPM3 TRPM8 TRPV1	Piezo
Adenosine	A${}_{1}$			Ca${}_{V}$							
	A${}_{2A}$									TRPA1	
Alanine	MrgD							K${}_{V}$7			
BAM 8-22	MRGPRX1									TRPV1	
Bradykinin	B${}_{1}$									TRPA1	
	B${}_{2}$		CaCC					K${}_{V}$7		TRPA1	Piezo2
										TRPM8	
										TRPV1	
CGRP	CGRP-R								Na${}_{V}$		
Endo-	CB${}_{1}$	ASIC	CaCC							TRPV1	
cannabinoids	CB${}_{2}$									TRPV1	
	CB${}_{?}$			Ca${}_{V}$							
Endothelin 1	ET${}_{A}$									TRPV1	
GABA	GABA${}_{B}$			Ca${}_{V}$							
H${}^{+}$	TDAG8/									TRPV1	
	(GPR65)										
Histamine	H${}_{1}$									TRPA1	
Neuromedin U	NMUR1						K${}_{V}$1.4				
							K${}_{V}$3.4				
							K${}_{V}$4				
Neuropeptide Y	Y${}_{2}$			Ca${}_{V}$							
Noradrenaline	α ${}_{2}$			Ca${}_{V}$							
	β			Ca${}_{V}$							
Nucleotides	P2Y${}_{1}$		CaCC	Ca${}_{V}$				K${}_{V}$7			
	P2Y${}_{2}$	ASIC	CaCC				K${}_{V}$1.4	K${}_{V}$7			Piezo2
							K${}_{V}$3.4				
							K${}_{V}$4				
	P2Y${}_{2/4/6?}$					K${}_{2}$P					
Opioids	MOP	ASIC		Ca${}_{V}$	GIRK	K${}_{2}$P			Na${}_{V}$	TRPV1	
	DOP		CaCC	Ca${}_{V}$							
	KOP			Ca${}_{V}$							
Prostaglandins	EP${}_{1}$									TRPV1	
	EP${}_{2}$								Na${}_{V}$	TRPV1	
	IP									TRPV1	
	FP					K${}_{2}$P					
Proteases	PAR1									TRPV1	
	PAR2	ASIC	CaCC							TRPV1	
	PAR4									TRPV1	
Serotonin	5-HT${}_{2}$	ASIC	CaCC							TRPV1	
	5-HT${}_{4}$					K${}_{2}$P			Na${}_{V}$	TRPV1	
	5-HT${}_{7}$									TRPV1	
	5-HT${}_{?}$			Ca${}_{V}$							
Somatostatin	SST${}_{4}$			Ca${}_{V}$							
Substance P/	NK${}_{1}$							K${}_{V}$7	Na${}_{V}$	TRPV1	
neurokinin A	NK${}_{2}$									TRPV1	
Urotensin	UTR						K${}_{V}$1.4				
							K${}_{V}$3.4				
							K${}_{V}$4				
?	MrgC			Ca${}_{V}$							

ASIC, acid sensing ion channel; CaCC, Ca${}^{2+}$-activated Cl${}^{-}$ channel; Ca${}_{V}$, voltage-gated Ca${}^{2+}$ channel; GIRK, G-protein activated; inwardly rectifying K${}^{+}$ channel; K${}_{2}$P, two-pore K${}^{+}$ channel; K${}_{V}$, voltage-gated K${}^{+}$ channel; Na${}_{V}$, voltage-gated Na${}^{+}$ channel; TRP, transient receptor potential channel; TRPA, ankyrin family; TRPM, melastatin family; TRPV, vanilloid family; BAM 8-22, bovine adrenal medulla peptide 8-22; CGRP, calcidonin-gene related peptide; CGRP-R, CGRP receptor; GABA, γ-amino butyric acid; ?, unknown.

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
