# Peer review of "Nociceptor Signalling through ion Channel Regulation via GPCRs"

_ijms, 2019, doi:10.3390/ijms20102488_

Round 1
Reviewer 1 Report
The authors have presented a very detailed, comprehensive review of the known nociceptors and how they are modulated by a wide variety of GPCRs. The manuscript is extremely well written and I only found a few very minor spelling/grammatical errors. My only complaint is that, as with all reviews of this scope, the text sometimes reads as a list of related pieces of information instead of a well-flowing narrative. Additionally, it would be nice if the authors delved a little more in depth about the clinical implications of GPCR modulation of nociceptors. Are any targets currently being investigated as a viable treatment for pain? What concerns do the authors have about targeting GPCRs given that it appears from their manuscript that many of them modulate a wide variety of downstream signaling pathways? Do the authors have any preclinical or clinical data that suggests one particular pathway might show particular promise?
Author Response
The authors have presented a very detailed, comprehensive review of the known nociceptors and how they are modulated by a wide variety of GPCRs. The manuscript is extremely well written and I only found a few very minor spelling/grammatical errors. My only complaint is that, as with all reviews of this scope, the text sometimes reads as a list of related pieces of information instead of a well-flowing narrative. Additionally, it would be nice if the authors delved a little more in depth about the clinical implications of GPCR modulation of nociceptors. Are any targets currently being investigated as a viable treatment for pain? What concerns do the authors have about targeting GPCRs given that it appears from their manuscript that many of them modulate a wide variety of downstream signaling pathways? Do the authors have any preclinical or clinical data that suggests one particular pathway might show particular promise?
Response: Thank you for the overall positive evaluation. We hope that we were able to detect all spelling/grammatical errors. With respect to clinical implications, we would prefer not to go any deeper into any kind of speculation, as we don’t have any experience regarding translational pharmacology.
Reviewer 2 Report
This is a well-written and comprehensive review to summarize the expression of ion channels in nociceptors and how specific GPCR signaling modulates these ion channel activity. The authors have done an excellent job in composing the manuscript. There are some minor points for the authors’ consideration.
1. It would be nice to have a summary table to show the 9 different ion channel families in nociceptors and the involved GPCR subtypes (and ligands).
2. While the focus of this review is on nociceptors, it might be helpful to emphasize on it and clearly define the nociceptor-related signaling. For example, TRPV1 is involved in pruriception and can be modulated by proton-sensing GPCR (e.g., TDAG8) and members of Mrgpr family.
3. Please check the expression of ASIC1a. To my knowledge, ASIC1a is predominantly expressed in small-medium sized DRG neurons, but not in large-sized DRG neurons.
4. Figure 7. Substance P can act on NK1R to enhance Kv7 activity via a G-protein-independent and tyrosine kinase-dependent pathway (PNAS 109: E76-E83, 2012; IJMS 20: 1596, 2019).
Author Response
This is a well-written and comprehensive review to summarize the expression of ion channels in nociceptors and how specific GPCR signaling modulates these ion channel activity. The authors have done an excellent job in composing the manuscript. There are some minor points for the authors’ consideration.
1. It would be nice to have a summary table to show the 9 different ion channel families in nociceptors and the involved GPCR subtypes (and ligands).
Response 1: Thank you for this suggestion. We have included such a table (table 2) and refer to it in line 663.
2. While the focus of this review is on nociceptors, it might be helpful to emphasize on it and clearly define the nociceptor-related signaling. For example, TRPV1 is involved in pruriception and can be modulated by proton-sensing GPCR (e.g., TDAG8) and members of Mrgpr family.
Response 2: In lines 86 to 87, we now mention that “TRPA1, TRPV1, TRPV3, TRPV4, TRPM8, as well as TRPC3 may contribute to the sensation of itch”. Furthermore, we include the modulation of TRPV1 via MRGPRX1 and TDAG8 in table 1
3. Please check the expression of ASIC1a. To my knowledge, ASIC1a is predominantly expressed in small-medium sized DRG neurons, but not in large-sized DRG neurons.
Response 3: We have rephrased the respective sentence (lines 247 to 248) and inserted an additional reference.
4. Figure 7. Substance P can act on NK1R to enhance Kv7 activity via a G-protein-independent and tyrosine kinase-dependent pathway (PNAS 109: E76-E83, 2012; IJMS 20: 1596, 2019).
Response 4: In lines 576 to 577, we now refer to the fact that “…this receptor may impinge on the functions of Kv7 channels through G protein-independent mechanisms as well…” and include the relevant reference.
Reviewer 3 Report
The review by Salzer et al provides a comprehensive discussion of the modulation of ion channels in primary nociceptor neurons by GPCR signaling. The attempt of the authors to give a comprehensive picture of the topic is laudable. The review is well written, and discusses an important topic, but I have some suggestions to improve it.
1. In line 297 the authors state: “the depolarizing mechanotransducer channel still needs to be identified”. This statement is incorrect, there is overwhelming evidence that Piezo2 channels are bona fide mechanically activated ion channels in DRG neurons, and they play roles in mechanosensory touch both in mice and in humans. Their mutations lead to human disease, and there are reports on their regulation by GPCR-s. They play a clear role in light touch, and there are ample data on their involvement in nociception as well, even though their exact role in the latter is still debated. While it is clear that there are other not yet identified excitatory mechanosensitive channels in DRG neurons, the authors should dedicate a section to Piezo2 channels.
2. The authors correctly point out that the roles of TRPV2, TRPV3 and TRPV4 are not physiological heat sensors. They mention the proposed role of TRPM3, and cite a review, but do not go into further discussion. I think TRPM3 deserves a more thorough discussion. Unlike the knockouts of TRPV2-4, the genetic deletion of TRPM3 leads to impaired noxious heat sensation (PMID: 21555074) and combined knockout of TRPV1, TRPM3, and TRPA1 leads to almost complete elimination of noxious heat sensation (PMID: 29539642). Furthermore, there are three recent publications demonstrating robust inhibition of TRPM3 channels in DRG neurons by a wide variety of Gi-coupled receptors (PMID: 28829742, PMID: 28826482, PMID: 28826490).
3. Line 89: The authors state: “TRPV1 channels and TRPM8 channels are never expressed on the same set of neurons.” While there may be some reports claiming this, there is ample data showing that in a subset of neurons both channels are expressed, for example: PMID: 2542006, PMID: 15050705, PMID: 27062607
4. Line 91: “all TRPA1 positive cells express TRPV1 as well [15]”. This statement is not correct either. While there is a substantial overlap between the two channels, it is not a complete overlap, there are plenty of TRPA1 positive that do not express TRPV1 and vice versa see for example: PMID: 29539642, PMID: 25420068
5. If the authors wish to be comprehensive, they may also consider a brief discussion on HCN channels. These channels are expressed in DRG neurons, they are modulated by cAMP and they have been implicated in nociception.
6. Line 39: ATP also act through P2Y receptors, which are GPCR-s
7. Line 167: “thought to contribute to the peripheral analgesic action of morphines” I think opioids would be a better word, instead of morphines
8. Line 170, the authors discuss NGF sensitization of TRPV1. They may consider mentioning that this happens via increased trafficking to the plasma membrane, as opposed to GPCR activation, which is a shift in the dose response to capsaicin, protons etc.
9. Line 277: “A family of ion channels that has an undisputed role in sensing noxious mechanical stimuli is the family of two-pore K+ channels”…. This is a somewhat confusing statement: my understanding is that mechanical stimuli activate these channels, which hyperpolarizes the neuron, thus they inhibit mechanotransduction, as the authors point out later. I would clarify this at the beginning, otherwise for readers outside the field this may become confusing, as from the opening statement they may think that these channel are inhibited by mechanical stimuli, which would make it possible for them to excite neurons in response to mechanical stimuli.
10. At the calcium activated chloride channel section, I would point out that unlike in the CNS, these channels were proposed to be excitatory in DRG neurons, due to the high chloride content of these cells (PMID: 20335661).
11. Line 401: “Only NaV1.6 to NaV1.9 channels can be found in nociceptive neurons.” Later in the same section: “The subunits NaV1.5, NaV1.8 and NaV1.9, which are expressed in nociceptive neurons” Please clarify which NaV channels are expressed in nociceptive neurons
12. Voltage gated Ca2+ channels: the authors may consider clarifying that some of the VGCC most likely function at the central process of DRG neurons, inhibiting neurotransmitter release.
13. Kv7 channels, the authors may mention the other names for these channels, KCNQ and M-current, to orient readers less familiar with these channels
Author Response
The review by Salzer et al provides a comprehensive discussion of the modulation of ion channels in primary nociceptor neurons by GPCR signaling. The attempt of the authors to give a comprehensive picture of the topic is laudable. The review is well written, and discusses an important topic, but I have some suggestions to improve it.
1. In line 297 the authors state: “the depolarizing mechanotransducer channel still needs to be identified”. This statement is incorrect, there is overwhelming evidence that Piezo2 channels are bona fide mechanically activated ion channels in DRG neurons, and they play roles in mechanosensory touch both in mice and in humans. Their mutations lead to human disease, and there are reports on their regulation by GPCR-s. They play a clear role in light touch, and there are ample data on their involvement in nociception as well, even though their exact role in the latter is still debated. While it is clear that there are other not yet identified excitatory mechanosensitive channels in DRG neurons, the authors should dedicate a section to Piezo2 channels.
Response 1: We now mention the role of Piezo channels in lines 325 to 330 and also include these channels in the introduction (line 72). Furthermore, we insert a new paragraph (2.3.2. GPCR regulation of Piezo channels) in lines 365 to 377.
2. The authors correctly point out that the roles of TRPV2, TRPV3 and TRPV4 are not physiological heat sensors. They mention the proposed role of TRPM3, and cite a review, but do not go into further discussion. I think TRPM3 deserves a more thorough discussion. Unlike the knockouts of TRPV2-4, the genetic deletion of TRPM3 leads to impaired noxious heat sensation (PMID: 21555074) and combined knockout of TRPV1, TRPM3, and TRPA1 leads to almost complete elimination of noxious heat sensation (PMID: 29539642). Furthermore, there are three recent publications demonstrating robust inhibition of TRPM3 channels in DRG neurons by a wide variety of Gi-coupled receptors (PMID: 28829742, PMID: 28826482, PMID: 28826490).
Response 2: We have included information on the expression pattern of TRPM3 channels in lines 96 to 99, functional information on these channels in lines 126 to 127 and a description of TRPM3 knock-out animals in lines 132 to 137. Furthermore, we have added a paragraph on Gi-mediated modulation of TRPM3 channels in lines 189 to 202 and include the references suggested by the reviewer.
3. Line 89: The authors state: “TRPV1 channels and TRPM8 channels are never expressed on the same set of neurons.” While there may be some reports claiming this, there is ample data showing that in a subset of neurons both channels are expressed, for example: PMID: 2542006, PMID: 15050705, PMID: 27062607
Response 3: We have rephrased the respective sentences (lines 88 to 94) and included the references suggested by the reviewer.
4. Line 91: “all TRPA1 positive cells express TRPV1 as well [15]”. This statement is not correct either. While there is a substantial overlap between the two channels, it is not a complete overlap, there are plenty of TRPA1 positive that do not express TRPV1 and vice versa see for example: PMID: 29539642, PMID: 25420068
Response 4: The according changes are contained in lines 88 to 94 (see above).
5. If the authors wish to be comprehensive, they may also consider a brief discussion on HCN channels. These channels are expressed in DRG neurons, they are modulated by cAMP and they have been implicated in nociception.
Response 5: As our entire manuscript deals with mediators of inflammatory pain, we would rather prefer not to include HCN channels which are believed to be relevant for neuropathic pain (and epileptic seizures; see e.g. Postea and Biel, Nat Rev Drug Discov. 10:903-14, 2011).
6. Line 39: ATP also act through P2Y receptors, which are GPCR-s
Response 6: In line 45, we now refer to the fact that ATP is an agonist at P2Y receptors.
7. Line 167: “thought to contribute to the peripheral analgesic action of morphines” I think opioids would be a better word, instead of morphines
Response 7: We have replaced “morphines” by “opioids” (line 181).
8. Line 170, the authors discuss NGF sensitization of TRPV1. They may consider mentioning that this happens via increased trafficking to the plasma membrane, as opposed to GPCR activation, which is a shift in the dose response to capsaicin, protons etc.
Response 8: Since NGF does not act via GPCRs, we would prefer not to add any further details.
9. Line 277: “A family of ion channels that has an undisputed role in sensing noxious mechanical stimuli is the family of two-pore K+ channels”…. This is a somewhat confusing statement: my understanding is that mechanical stimuli activate these channels, which hyperpolarizes the neuron, thus they inhibit mechanotransduction, as the authors point out later. I would clarify this at the beginning, otherwise for readers outside the field this may become confusing, as from the opening statement they may think that these channel are inhibited by mechanical stimuli, which would make it possible for them to excite neurons in response to mechanical stimuli.
Response 9: We have rephrased the relevant sentence which now reads: “One family of ion channels that contribute to the sensation of noxious mechanical stimuli is the family of two-pore K+ channels” (lines 306 to 307).
10. At the calcium activated chloride channel section, I would point out that unlike in the CNS, these channels were proposed to be excitatory in DRG neurons, due to the high chloride content of these cells (PMID: 20335661).
Response 10: In lines 412 to 414, we now explain that “In dorsal root ganglion neurons, the inflammatory mediator bradykinin was demonstrated to increases neuronal excitability via gating of CaCCs which leads to a depolarizing Cl- efflux due to comparably high intracellular Cl- concentrations…”.
11. Line 401: “Only NaV1.6 to NaV1.9 channels can be found in nociceptive neurons.” Later in the same section: “The subunits NaV1.5, NaV1.8 and NaV1.9, which are expressed in nociceptive neurons” Please clarify which NaV channels are expressed in nociceptive neurons
Response 11: By rephrasing the relevant sentence (lines 458 to 460) we have eliminated this contradiction.
12. Voltage gated Ca2+ channels: the authors may consider clarifying that some of the VGCC most likely function at the central process of DRG neurons, inhibiting neurotransmitter release.
Response 12: We now allude to the fact that “…inhibition of CaV2.x channels leads to reduced Ca2+ influx and concomitantly reduced transmitter release from the peripheral nociceptive neurons onto second order neurons of the pain pathway located in the spinal dorsal horn” (lines 518 to 521).
13. Kv7 channels, the authors may mention the other names for these channels, KCNQ and M-current, to orient readers less familiar with these channels
Response 13: The respective sentences now read as follows: “…with five known members (KV7.1- KV7.5) encoded by KCNQ1-5 genes [209]. Four monomers come together in a homo-or heterotetrameric configuration in a subunit-specific way to yield a functional KV7 channel [215]. The electrophysiological correlate of KV7 channel activity is a slowly deactivating, non-inactivating current that has an activation threshold below -60 mV. This conductance is also known as M current as it was originally described as a current that is suppressed by an activation of muscarinic acetylcholine receptors [216]” (lines 557 to 562).